# Interdigitated Organic Sensor in Multimodal Facemask’s Barrier Integrity and Wearer’s Respiration Monitoring

**DOI:** 10.3390/bios12050305

**Published:** 2022-05-06

**Authors:** Marina Galliani, Laura M. Ferrari, Esma Ismailova

**Affiliations:** 1Mines Saint-Etienne, Centre CMP, Département BEL, 13541 Gardanne, France; marina.galliani@emse.fr; 2INRIA, Université Côte d’Azur, 06902 Sophia Antipolis, France; laura.ferrari@inria.fr

**Keywords:** e-mask, ink jet, PEDOT:PSS, organic sensor, wearable monitoring, COVID-19

## Abstract

Facemasks are used as a personal protective equipment in medical services. They became compulsory during the recent COVID-19 pandemic at large. Their barrier effectiveness during various daily activities over time has been the subject of much debate. We propose the fabrication of an organic sensor to monitor the integrity of surgical masks to ensure individuals’ health and safety during their use. Inkjet printing of an interdigitated conducting polymer-based sensor on the inner layer of the mask proved to be an efficient and direct fabrication process to rapidly reach the end user. The sensor’s integration happens without hampering the mask functionality and preserving its original air permeability. Its resistive response to humidity accumulation allows it to monitor the mask’s wetting in use, providing a quantified way to track its barrier integrity and assist in its management. Additionally, it detects the user’s respiration rate as a capacitive response to the exhaled humidity, essential in identifying breathing difficulties or a sign of an infection. Respiration evaluations during daily activities show outstanding performance in relation to unspecific motion artifacts and breathing resolution. This e-mask yields an integrated solution for home-based individual monitoring and an advanced protective equipment for healthcare professionals.

## 1. Introduction

During the recent COVID-19 pandemic outbreak, the entire world faced multilevel healthcare management challenges. Delays in medical care, with canceled consultations or postponed surgeries, were often experienced due to the overloading of hospitals and paralyzing whole medical services. The mortality risks at emergencies have been increasing for both SARS-CoV-2 (severe acute respiratory syndrome coronavirus 2) related and unrelated patients [1]. In such a context, the effective strategies to prevent the disease spreading, to perform early detection, fast screening, and to remotely monitor infected patients became an urgent need. Mobile technological solutions have played a key role in multiple aspects, from contact tracing to the monitoring of physiological metrics, facilitating personal health training during the pandemic [2]. Wearing of facemasks in public spaces as a proper protection measure to mitigate the virus spreading has been generally imposed. The physical barrier created by the facemask hinders the scattering of water droplets and aerosols containing the virus. Even though the health authorities have acknowledged the benefits of facemasks, controversies about their proper management and disposal quickly arose. The WHO recognized the need to change the mask immediately if it became damp [3]. Without any quantitative tools to properly define the barrier properties of the mask, first guidelines were unclear for the population and the use of the masks has not been following secure protocols. The status of the barrier integrity alertness of such protective equipment presents a great prospect to develop a good and safe usage practice, particularly in the general public, and to indicate when the mask needs to be changed. From a technological point of view, wearable facemasks are suitable for smart consumer-grade devices development. Combining facemasks with sensing devices allows not only to monitor the facemask’s status but can also track proximate health risks parameters. In this respect, electronic masks (e-masks) are foreseen as promising tools for a variety of on-body sensing in related or unrelated pandemic scenarios [4].

Through the actual pandemic, an international effort has been made towards the e-mask’s development. The wearable devices examples have been largely reported in measuring physiological constants as a SARS-CoV-2 early-detection tool [5]. Y. Wang et al. have shown the fabrication of a flexible humidity sensor on a non-woven fabric to monitor humidity and respiration using graphene oxide [6]. Similarly, Y. Pang et al. developed a humidity sensor for respiration, speaking, and whistle rhythm monitoring from a conductive porous graphene network [7]. A moisture-sensitive RFID antenna has been placed inside a facemask to discriminate a slightly damp mask from a wet one [8]. Wang et al. [9] have developed a portable respiratory moisture and 3D optoelectronic sensor through the integration of a PEDOT:PSS-coated fiber array into a facemask. Wireless CO_2_ monitoring through an opto-chemical sensor combined with a flexible, battery-less, near-field-enabled tag was fabricated on a facemask by P. Escobedo et al. [10]. These technological developments have shown good performances and facilitated the demonstration of multiple proofs of concept in creating e-masks. To reach the final user and enable availability of such devices on a large scale, rapid, cost-effective fabrication and disposable solutions need to be further foreseen. Actual examples require the sensors to be fabricated onto a substrate and then added into the mask, generally by sticking it in a certain mask section. This approach makes the sensor a bulky accessory that alters the facemask’s conformability, thus interfering with its normal filtering properties and the user’s respiration process. Moreover, a sensor made in this way risks detaching from the mask and disturbing its function by movement artifacts. Additionally, the fabrication processes, that need several steps and multiple materials, will require complex technological adjustments at the masks’ manufacturing facilities and therefore limit their industrial transfer. Therefore, the key challenge in the fabrication of the e-mask, able to be rapidly available on the market, is the development of an integrated (in respect to the masks’ manufacturing flow) and unobtrusive sensor, in respect to the mask’s initial usage specifications, by applying fast and large-scale cost-efficient fabrication processes.

Among multiple functional materials, conducting polymers (CPs) are well-positioned in the field of flexible and wearable electronics. The poly(3,4-ethylenedioxythiophene) (PEDOT)-based derivatives are today the most employed ones due to their soft nature, cost-effectiveness through their commercial availability, versatile processability, sustainability and compatibility with green processing, particularly targeted in biomedicine [11]. The most popular version is a commercially available water suspension of PEDOT with PSS, polystyrene sulfonate. Its adoption for the fabrication of biomedical devices extends from implantable microelectrode [12,13], cutaneous surface electrophysiology recording electrodes [14], to transistors for biochemical sensing [15,16]. The inkjet printing of PEDOT:PSS is a digital and fast prototyping technique that stands out for its additive processing, non-contact patterning, low-cost, high resolution, down scalability, and mild processing conditions. Demonstrations of the inkjet printing on wearable substrates led to the conception of organic bioelectronic sensors with remarkable performances. Using the printing technique, the functional inks are jetted on the substrates in small amounts, creating thin conformable films and resulting in fully integrated and flexible sensors [17,18,19,20,21].

Herein, we present one-step low-cost fabrication of a fully polymeric inkjet-printed sensor on a surgical facemask. Using the solution processable printing offers large-scale fabrication and roll-to-roll process transfer that is compatible with industrial non-woven textiles manufacturing and mask production. The introduction of printed interdigitated electrode patterns on the inner layer of the mask requires only a particular attention at the mask assembly step. This e-mask shows dual-sensing capabilities based on the printing resolution characteristics and interdigitated electrodes design study. The achieved dampness and respiration rate monitoring follow the resistive and capacitive sensing mechanisms, respectively. We not only detect the wet status of the mask, but also demonstrate the capability of the sensor to observe and explain the water droplets interaction with the mask through an electronic signal in various relative humidity and temperature conditions. Additionally, we monitor the user’s respiration rate (RR), which is considered a vital parameter to identify breathlessness or difficulty breathing, both symptoms that might be correlated with respiratory tract infections such as in COVID-19 disease [22,23]. Respiration detection with the e-mask precisely follows the breathing phases during standing, walking, and running in comparison with a commercial chest belt monitoring system. The mask wetting process happens during physical activities, and it reaches a damp state during running. The wetting process is reversible since the sensor and the mask remain permeable to the air and the evaporation occurs continuously. This does not affect the respiration quality detection, which is shown to be higher due to the capacitive response of the sensor. The integrated PEDOT:PSS sensor, connected to a portable acquisition system, represents a wearable platform that is extremely beneficial for individual ubiquitous health monitoring and for frontline healthcare personnel protection, particularly during a pandemic framework [24].

## 2. Materials and Methods

### 2.1. Materials

Aqueous PEDOT:PSS dispersion (Clevios PH1000) was purchased from Heraeus Deutschland GmbH. The ink formulating chemicals DMSO, IPA, and TWEEN 20 were purchased from Merck. Commercially available surgical facemasks were used for the e-mask prototype manufacturing.

### 2.2. E-Mask Fabrication

To fabricate the e-mask, the PEDOT:PSS commercial formulation was adjusted to be inkjet printable and to enhance its electrical performance. The Clevios PH100 was mixed with DMSO (10% *w*/*w*), IPA (5% *w*/*w*), and a nonionic surfactant TWEEN20 largely used in biochemical applications. Figure 1 (left) schematizes the ink preparation through the addition of solvents, a surfactant, and a crosslinker. Prior to printing, the ink was filtered with a 0.2 μm cellulose acetate syringe filter to remove the biggest polymeric particles that might clog the printhead nozzles. Dimatix DMP-2800 (Fujifilm Dimatix, Inc., Santa Clara, CA, USA) inkjet printer with a 10 pl cartridge was used for printing. Two interdigitated electrodes (IDEs) were directly printed with PEDOT:PSS ink onto the inner filtering layer of the facemask, detached from the tree layers stack of the mask. Electrical properties in function to the number of printed layers are summarized in Appendix A. The facemask inner layer was taped on a plastic foil to facilitate its handling. The designed IDEs consist of 2 electrodes with 6 interdigitated fingers each. The printed sensor, made of 6 consecutively deposited layers, was cured in an oven at 60 °C for 1 h. Figure 1a is a schematic of the fabrication route and the sensor design.

As the portable acquisition system, a sensor kit purchased from PLUX Wireless Biosignals S.A. was used as a data acquisition board. We used the electrical conductance module connected to a Micro Controller Unit (MCU), a battery, and a low-energy Bluetooth module for data transfer. The IDEs were electrically connected to the two lead cables of the acquisition module and the data was processed in real-time via Bluetooth on a personal computer.

### 2.3. E-Mask Characterization

#### 2.3.1. E-Mask Response to an Aerosol Flow

An aerosol flow was used to determine the e-mask response to water content. A portable aerosol spray for the treatment of the respiratory tract was employed in the sensor calibration. The e-mask was placed inside of a closed plastic box with an inlet for the aerosol flow. This allowed the relative humidity to reach 100% and to obtain water condensation and consecutive drop formation. The aerosol outlet was placed 3 cm in front of the sensor’s sensing area. The progressive wetting of the mask was obtained with the aerosol flow of 300 μL/min. After 6 min, the flow was stopped, and the sensor response was recorded for an additional 1 min.

#### 2.3.2. E-Mask Response to Relative Humidity Variations

The e-mask calibration in a controlled relative humidity (RH) was performed in a Temperature/Climatic Stress Screening Chamber (CTS, GmbH). The e-mask was placed inside the humidity chamber with a waterproof-protected electronics module. The temperature was kept constant at 30 °C and the RH ramped from 60% to 95% to simulate the facemask conditions that normally occur during daily use [25].

#### 2.3.3. IDEs Response to the Temperature Variation

PEDOT:PSS printed lines 0.5, 1, and 2 mm wide and 25 mm long were placed on a hot plate to assess the IDEs electrical behavior in respect of the temperature fluctuation. While increasing the temperature from 20 to 50 °C, the electrical resistance was measured with a digital multimeter. The normalized electrical resistance change (%) was then calculated in relation to the temperature over 3 experimental measures.

#### 2.3.4. Contact Angle Measurement

The contact angle measurements were performed with an optical contact angle measuring unit (OCA200, Apollo Instruments, Compiegne, France). To investigate the wetting properties of the facemask filter, contact angle measurements were performed in controlled humidity at room temperature. A dry facemask filter specimen was placed on the measurement sample holder and a 20 μL water drop was dispensed on the filter. A photograph time lapse was recorded with a digital camera. The contact angle was measured through a drop shape analysis. For the damp filter study, the same measurement was performed on the filter specimen that was submerged in water for 10 s. The filter was then recollected, and the excess of water was removed before placing it on the sample holder.

### 2.4. E-Mask Evaluation

This study was performed strictly for technological demonstration and no personal data were collected from participants. Informed consent was obtained from two participants in agreement with local ethical regulation. To assess the e-mask performances, we asked a participant to wear the e-mask and a commercial chest belt was equipped with an inductive respiration sensor for comparison. The participant wore the mask and the chest belt, positioned around the thorax, during standing, walking, and running activities. Eventually, the participant was asked to remove the e-mask to study the e-mask response when unworn. The signal was acquired by the portable acquisition unit and wirelessly transmitted to the personal computer, as detailed above.

### 2.5. E-Mask Data Processing

The acquired time series data from the IDEs (μS) and the chest belt (V) were collected in real-time through the OpenSignals software 2.1.1 (PLUX Wireless Biosignals, Lisbon, Portugal). The data were manually annotated in relation to the performed activity and representative data were illustrated in figures. The e-mask response towards the wetting process was calculated by sampling the raw data each 100 s. The peak detection analysis for the breathing rate analysis was carried out with the Matlab local maxima function from raw data. The breathing rate was calculated by dividing the number of peaks by the recording time expressed in minutes.

## 3. Results and Discussion

The following section describes the sensor characteristics and working principles in correlation to the adopted design. Next, it presents the e-mask multimodal sensing and motion artifacts’ tolerance.

### 3.1. IDEs Printing on Facemask

The fabrication of PEDOT:PSS-interdigitated sensor on the mask is shown in Figure 1. To guarantee its direct and monolithic integration, the sensor is printed onto the internal filter layer of the commercial medical mask (Figure 1a), which is made of a non-woven synthetic fabric. The arrangement of the ink on the fabric layer can be appreciated in Figure 1b, which shows a cross-section view of PEDOT:PSS printed line. The printing resolution on such a surface is evaluated by a resolution pattern printing with different line dimensions. Figure 1c shows achievable printing of PEDOT:PSS lines ranging from 500 to 40 μm. The wicking properties of the non-woven fabric enable low spreading of the deposited ink drops. Therefore, a well-defined pattern can be obtained with sharp borders, showing high printing quality with maximum resolution as low as 40 μm. The electrical properties of the lines are defined in function of printed layers number. The electrical resistance (R) of rectangular patterns (shown in Appendix A) with 1, 3, 6, and 9 consecutively printed layers is reported in Appendix A. As expected, the R linearly decreases with the increasing number of printed layers, from 120 ± 60 kΩ (1 layer) to 2.3 ± 1.6 kΩ (9 layers) over a centimeter line. The final IDEs were printed with 6 layers since it minimizes material’s use and enables a fast process and good conductivity (see Appendix A), resulting in a great deposition homogeneity of the lines, thus with stable conductive paths and through a quick fabrication process. The IDEs’ sensing area is placed in the center of the mask and defined by the interdigitated fingers with up to 0.52 mm interelectrode distance. The electrodes are then coupled by two longer perpendicular connection lines drawn to the exterior of the mask towards the oval-shaped connection pads (Figure 1d).

### 3.2. E-Mask Design and Working Principle

The idea of IDEs design allows for increasing the effective surface area of sensing without increasing the electrodes’ dimensions. The geometry considerably affects the sensor’s performance and its sensitivity. The typical design features are the number of fingers (n), the length (L) and the width of the electrodes (W), and the gap distance (G) between the electrodes, which allows tuning the sensor’s sensitivity [26]. The specifications of the sensor design used in this work follows the sensor dimensions study (Appendix A). Three electrodes with different W and G were fabricated. The W dimension was chosen to allow IDEs sensing area to cover a major portion of the mask in the mouth area. The choice of G was set by considering the average diameter of water droplets deposited on the sensor during the dampness process. The W equal to 1 mm and the G equal to 0.5 mm were found to be the optimal values for our application. Twelve fingers (n) with a length (L) of 18 mm are needed to position the sensitive area in front of the mouth and enable the sensor output connection at the mask end. These dimensions permit efficiently covering the area of interest related to the user’s breathing while ensuring a good sensor sensitivity.

IDEs-based sensors can have a capacitive, resistive, or mixed (e.g., impedimetric) working principle, according to their design. In our e-mask, IDEs show a resistive behavior for the dampness monitoring, where water drops create a short circuit between two IDEs. A capacitive behavior is dominant in the respiration monitoring where breathing vapors alter the dielectric properties of the substrate between IDEs. These two different mechanisms, described in detail below, occur simultaneously and allow the exceptional multimodal sensing of the e-mask.

#### 3.2.1. Dampness Sensing Mechanism

This sensing mechanism is related to the monitoring of the mask wetting process because of a progressive water accumulation from condensed aerosol droplets. In the center of the mask, where the IDE is printed, the water droplets reach the filter layer and coalesce in larger drops driven by the water surface tension and the substrate’s surface energy properties. In this case, the e-mask works as a resistive sensor, where the water drops cause the electrical bridging between IDEs fingers. Thus, the sensor passes from an open to a closed circuit. This electrical circuit is triggered by a variable conductance value. Indeed, statistically newly deposited droplets bring additional electrical connections between the electrodes over the interdigitated surface area. Therefore, the sensor monitors the deposited water amount, which echoes the mask dampness by following a progressing conductance increase over time.

#### 3.2.2. Respiration Sensing Mechanism

This sensing mechanism is related to the respiration monitoring as the detection of the RH local changes, which is caused by the humid outbreath. In this case, the IDEs work as a capacitive sensor in which the capacitor’s dielectric constant corresponds to the temporal status of the facemask filter material [27,28,29,30]. In this case, when the humid air passes across the e-mask, it shortly increases the filter’s dielectric constant. This change of capacitance value is indirectly quantified from a change of the conductance between the IDEs. Here, the mask remains dry and the resistive element of the circuit, presented in Section 3.2.1, is negligible.

### 3.3. E-Mask Characteristics

Tracking the facemask wetting allows control of the facemask filtering efficiency and barrier integrity. Facemask filters are made of nonabsorbent polymeric fibers, which are electrostatically charged to improve their airstream particles capture. It has been demonstrated that their filtration performance decreases as humidity increases, because the water amount neutralizes the fibers’ electrostatic charges [31]. Therefore, a damp mask shows reduced filtration capabilities. First, we characterize the e-mask capability to detect water droplet accumulation at its surface. Figure 2a shows the sensor calibration curve with respect to the amount of condensed water using a water aerosol spray with 300 μL/min characteristics. The red dashed line represents a linear fit (R_2_ = 0.99) that validates the linear electrical response. The linearity indicates constant sensing capabilities over the range of interest and allows to correlate consistently the electrical output signal to the water amount. This behavior rigorously correlates sensor’s electrical conductance (μS) to a specific wetting status. These data were further extrapolated for the sensor’s time response, reported in the bottom right inset of Figure 2a. The time response shows the conductance value raises from 0 to 1 μS in one minute and reaches a maximum value of 3 ± 0.2 μS between 4 to 6 min and then decreases. This maximum conductance corresponds to 1800 μL of water condensate, according to the aerosol rate specifics, and indicates the sensor is saturated in water. Additionally, the top-left inset of Figure 2a shows the sensor’s photograph at 4 min of exposure to the water condensation. The visual inspection of the e-mask shows the water droplets, which partially damp the sensor that, in response, results in a sensor output value of 2.45 ± 0.47 μS. Hence, we considered 3 ± 0.2 μS as the threshold value to distinguish a barely wet mask from a damp one, and at which the sensor should alert the user to change the mask. The wetting mechanism is moreover characterized by a variation in the filter’s surface properties. When the facemask filter layer is dry, it shows water-repellent properties. This is visible in Appendix A, showing a water drop on the filter surface with a contact angle of 136° which does not change during the time. The filter layer maintains its hydrophobic surface. On the contrary, we observe the progressive water drop absorption on the initially damp filter (Appendix A). The drop absorption and spreading are displayed by the progressive evolution of the contact angle which decreases from 132°, when the drop is deposited onto the filter (0 s), to 0°, when it is completely absorbed (14 s). We believe that when a certain water amount saturates the filter surface, deposited water drops are pulled within the filter fibers by the capillary forces. Therefore, the water-repellent properties are lost and a water communicating channel is established, which impairs the mask protecting function. This scenario can happen when an extensive amount of water droplets appear during heavy breathing. During a normal respiration, only a small amount of the droplets can condense on the hydrophobic surface, which dries over time. This rationale can be applied to explain the overshoot in the time response in the Figure 2a after maximum conductance is reached. Indeed, the mask is an air permeable support, and the water droplets’ evaporation can be naturally envisioned as well as water evaporation in a damp mask. The sensor’s response, which is observed around 6 min, takes place thanks to the subtle balance between the simultaneous water accumulation and its evaporation over the whole surface of the IDEs sensing area.

We then investigated the sensor’s response to RH variations that occurred during breathing cycles. We consider that the variation of the RH during outbreath ranges between 75–95% [25]. For this evaluation, the e-mask was assessed at a constant temperature of 30 °C and the RH was monotonically increased from 60% to 95% (Figure 2b). At 60% RH, corresponding to the typical humidity of the inhale [25], the IDEs show 0 μS as the output response. When the RH increases, the sensor output changes while reaching the maximum value of 0.07 μS at 95% RH (Figure 2b). Therefore, the resulting breathing humidity does not interfere with the dampness measure as the water accumulation only starts above 1 and up to 3 μS.

Considering that human breathing involves temperature variations between 28–33 °C [25], we investigated the sensor’s electrical stability in respect to the temperature change. Figure 2c shows the electrical resistance of PEDOT:PSS-printed lines in the temperature range of 20° to 50 °C. The resistance fluctuation over this temperature window does not exceed 8%. In the range of 25 to 35 °C, the variation is only ~1 ± 0.71%. This insignificant temperature dependence reveals the sensor’s stability to the wearer’s body temperature as well as to the environmental temperature changes.

### 3.4. Multimodal E-Mask System Validation

After assembling the mask together with the printed IDEs sensors, its performance was evaluated in daily usage conditions. A volunteer wore the e-mask in a static position, representing standing, and during physical activities, walking, and running. Figure 3a shows the final device allowing us to appreciate the seamless integration of the printed sensor into the medical mask. The combination with a commercial portable miniaturized acquisition module ensures the imperceptible wearability of the whole system. The IDEs connection pads are located on the side, where the electrical connection to the electronic module is less noticeable to the wearer as well as impactful to the sensor’s steadiness.

#### 3.4.1. E-Mask Dampness Monitoring

The first evaluations of the IDEs e-mask sensor in use during daily activity is provided in Figure 3b. The figure presents the sensor’s time response to standing and walking for about 30–35 min from two distinct masks. This condition mimics regular short-term facemask usage. The acquired signals, on average ranging around 1 ± 0.44 μS, show an increase of the sensor’s wetting after 5–10 min of wearing, followed by a gradual decrease and stabilization. In both mask cases, the maximum signal did not reach the dampness threshold of 3 μS, defined from the calibration data in Figure 2a. The decrease and stabilization of the wetting process indicate that the evaporation process also occurs. Hence, the sensor does not hinder the normal air permeability of the facemask, which is crucial in its filtration capability and proper functioning [31]. When the facemask is removed (delimited by the dashed line in Figure 3b), the sensor output signal quickly decreases towards zero μS. At this moment, a rapid evaporation of the accumulated humidity is taking place, which leads to the mask’s dry status restoration. In both recordings, the signals drop to values <0.13 μS in less than 4 min. Additional data provided in Appendix A demonstrates that in one minute the output signal is almost halved after the mask is removed from the subject’s face. This fast recovery time of the air permeable IDEs sensor points to the intrinsic drying facemask properties.

The further investigation of the e-mask wetting process was observed during a moderate physical exercise. It is known that during an exercise while wearing a mask, a higher RR causes a superior humidity condensation and saliva droplets discharge together with the presence of face-sweat. During a running activity (Figure 3c), the average sensor output response is higher (1.97 μS) than in early reported walking activity (1.00 μS). The e-mask undergoes phases of high amplitude peak responses after 7 min of exercise (at minutes 8, 9.3, and 12.3). It is likely that these peaks are the result of saliva droplets and face-sweat deposition onto the sensor, corresponding to the saturation of the dampness threshold 3 μS, defined in Figure 2a. The peaks are then followed by a progressive 5 min signal decrease before the next phase of the saturation. This behavior is a result of the simultaneous water accumulation over the whole sensor area and evaporation described in Section 3.3, which is promoted by the sensor’s and the mask’s air permeability. These results confirm the capability of the e-mask to monitor its continuous wetting processes occurring during exercises. They also reveal that a damp mask is capable of restoring its initial barrier properties due to the persistent water evaporation.

#### 3.4.2. Respiration Rate Monitoring

The overall e-mask time response consists of periodic peaks with an amplitude of 0.22 ± 0.01 μS and a baseline drift around 0.01 μS (Figure 3d). This signal was compared to recordings from an inductive respiration sensor of a chest belt during a slow and deep respiration. It monitors the overall displacement of the thorax through a piezoelectric element. One breathing cycle is identified by a pattern with maximum and minimum voltage peaks. The time value between two consecutive peaks is used to calculate the RR. As shown in Figure 3d, sensors’ output comparison reveals a temporal activity superposition between both detectors. Although the sensing principles are different between the two sensors, the output signals allow to similarly monitor and reveal the breathing cycle’s characteristics. Since the e-mask records a local and temporary RH variation, the outbreath corresponds to the RH signal increase, while during the inhale the signal drops to the baseline value. The observed sensor response that returns back to the baseline value after each exhale confirms that the presented e-mask is air- and vapor-permissive and does not hinder the facemask gas permeability. Therefore, the time between peaks is used to estimate the breathing frequency. The peak amplitude here seems to correspond to the strength of the respiration and the width defines its length. Further studies are necessary to conclude if the sensor can be used for the breathing quality assessment.

#### 3.4.3. Motion Artifacts Tolerance

The body motion artifacts can significantly impair wearable sensors’ performances. A comparison of the respiration cycles acquired with the e-mask and from a commercial chest belt are shown in Figure 3f. The chest belts are broadly used as commercially available wearable solutions in monitoring respiration and heart rates during exercises. Two-time responses for the e-mask (μS) and the belt (V) are simultaneously recorded over 80 s. The participant was first asked to statically stand (first 35 s) and then to start running in place. This activity transition results in an increase of high amplitude noise level in both recordings. In the chest belt signal, the noise considerably hides the respiration pattern and requires an advanced data processing to estimate the RR values. In the e-mask output signal, the noise content is visibly reduced even during running. The typical breathing pattern is well-defined in the raw signal. Appendix A shows a great accuracy of the RR estimation during relaxing and running by comparing the automatic peaks detection in the raw signal with and without presence of the motion-induced noise. These observations highlight the remarkable recording quality with the e-mask thanks to the IDEs sensor’s ubiquitous integration and its lightweight. Additionally, using IDEs for monitoring RR is beneficial as the output signal is a measure of conductance and not voltage, which is shown to be less sensitive to unspecific noises.

#### 3.4.4. Multimodal Monitoring

The working principle of the IDEs in detection of the local humidity variation and dampness have resulted in both the user’s breathing and the facemask dampness monitoring in different usage scenarios. To assess the sensors’ resolving capability over different breathing frequencies, the RR peaks detection was computed when the subject was standing, walking, and running. As depicted in Figure 3f, the number of peaks in a 20 s time window increases with the escalation of the physical activity, providing an estimated RR of 8.40, 13.01, 17.54 breaths/min, respectively. The e-mask exhibits an excellent capability of monitoring deep, normal, and fast breathing, covering all daily activity scenarios.

From the electrical circuit point of view, the dampness and respiration monitoring involve different sensing processes. In Figure 3f, during the running, a clear distinction of the breathing pattern is observed while the sensor is considered wet, as its response shows values that are higher than the dampness threshold. The respiration pattern is also visible in the e-mask response while it is in an intermediate wetting and drying state (Appendix A). The time responses of the e-mask when the conductance is increasing due to its wetting contains waveforms of the respiration. The same pattern is noted during the mask drying over the decreasing conductance signal. Such multimodal monitoring with the e-mask implies that the wetting status and RR tracking occur independently.

## 4. Conclusions

In this study, we have demonstrated the fabrication of a ubiquitous sensor on a protective facemask using scalable processing and allowing a fast and low-cost production. Using conducting polymers as an active material of the IDEs and inkjet printing as a high resolution deposition process on the inner filter layer ensures consistent air exchange across the facemask layers without any physical interference from the sensor at use. The sensor’s linear electrical response to the aerosol condensate has revealed its capability for the mask’s dampness monitoring. Simultaneously, this e-mask has an excellent sensitivity to RH variations used for the monitoring of human breathing cycles at low, normal, and fast rates. Moreover, the presented multimodal e-mask shows improved stability towards motion artifacts, providing an accurate detection during daily use.

Considering that a miniature portable acquisition system can be plugged and reused (Appendix A), the e-mask can be fully disposable with a low environmental impact. Future work can be extended for the evaluation of an easy translation of PEDOT:PSS IDEs to other types of protective equipment, such as N95 or FFP respirator masks, for smart consumer-grade device development.

## Figures and Tables

**Figure 1 biosensors-12-00305-f001:**
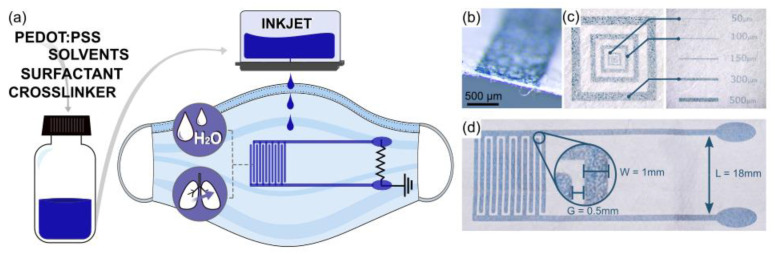
E-mask fabrication process. (**a**) Schematics of the e-mask fabrication route composed of the printable PEDOT:PSS ink formulation (on the left), the sensor direct printing onto the medical facemask (on the right), and the possibility of dual monitoring with IDEs. (**b**) Picture of a printed IDEs finger, a cross-section view. (**c**) Optical images of the square resolution pattern printed on the facemask filter fabric with a zoomed view of the printed lines with different width. (**d**) Picture of the printed IDEs sensor with the specific dimensions: the length between the electrodes (L), the width of the electrodes’ fingers (W), and the gap between fingers (G).

**Figure 2 biosensors-12-00305-f002:**
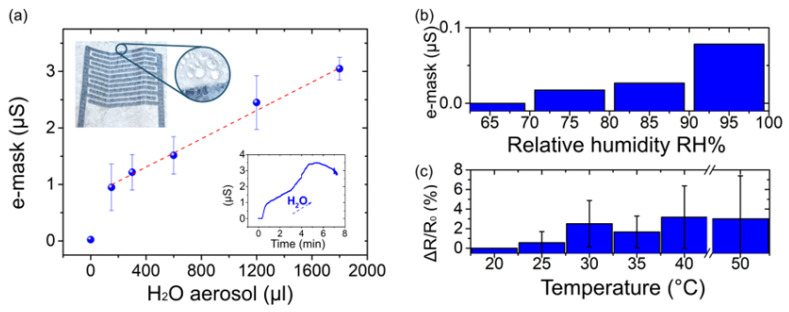
E-mask characterization. (**a**) E-mask calibration curve with respect to H_2_O aerosol flow. Error bars represent the standard error over three measurements; the dotted red line is the linear fit. Top-left inset shows the damp sensor with the zoom on the condensed water droplets formation. Bottom-right inset shows the e-mask time response to aerosol flow (rate of 300 μL/min). (**b**) E-mask signal output (μS) variation with respect to the relative humidity (RH varies from 65% to 95%). (**c**) Electrical resistance variation (%) of printed PEDOT:PSS lines with respect to the temperature increase (from 20 °C to 50 °C).

**Figure 3 biosensors-12-00305-f003:**
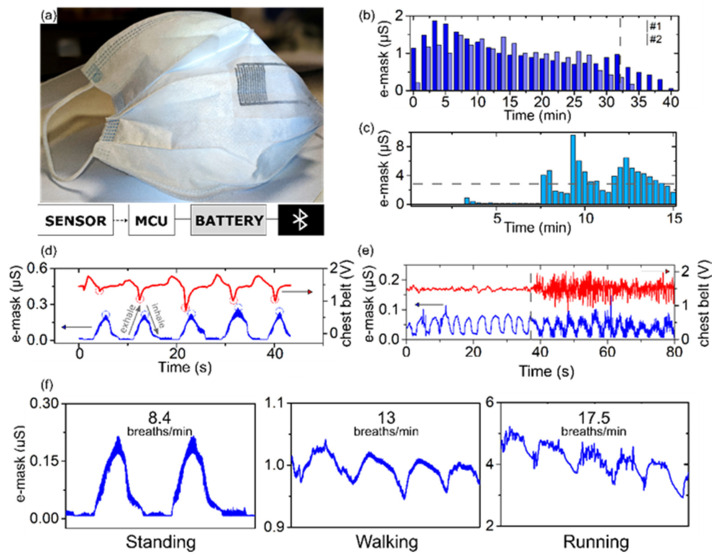
E-mask system response: (**a**) Picture of the final e-mask with printed IDEs (dark lines). On the bottom, a schematic presentation of the portable acquisition system for the data collection, the digitalization, and the wireless signal transfer. (**b**) E-mask time response of the wetting process monitoring during a walking activity (data from 2 different masks). The dashed line indicates when the e-mask is removed and left to dry at ambient conditions. (**c**) E-mask time response of the wetting process monitoring during running activity. The dashed line reports the 3 μS threshold associated with the facemask damp condition. (**d**) Time window of the respiration rate monitoring: the e-mask (blue) and chest belt (red). Small circles indicate respiration cycles. (**e**) Raw signals of the e-mask (blue) and chest belt (red). The vertical dashed line indicates the passage from resting to running. (**f**) Estimation of the respiration rate from the e-mask raw signal at standing, walking, and running activities. RR estimation is given over a time window of 20 s and sensor’s response below and over the dampness threshold, demonstrating a possibility for multimodal detection in running conditions.

## Data Availability

Data available on request from the authors.

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
