# Peer review of "Interdigitated Organic Sensor in Multimodal Facemask’s Barrier Integrity and Wearer’s Respiration Monitoring"

_biosensors, 2022, doi:10.3390/bios12050305_

Round 1

Reviewer 1 Report

see attached file

Author Response

Please see our point by point response in the doc.

Reviewer 2 Report

In this manuscript, an organic sensor to monitor the integrity of surgical masks was proposed and experimentally verified. This idea is novel and the results are good. However, there are still some major problems. The specific comments are as follows.

  1. Computers are required for the data processing. As we can not look at the computers all the time, how can people be notified immediately that their facial masks can no longer be used?
  2. Batteries and MCU are also needed. Are they on the masks as well? Figures of them should be presented.
  3. How much is the cost of this kind of E-mask? Usually, the masks are disposable and discarded after use. Is it worthwhile to add this sensor on the mask?
  4. According to Figure 3(b), when the e-mask is removed and left to dry, the sensor output signal quickly decreases. Does that mean this e-mask can still be used even after the signal has been greater than 3 uS? How to determine whether this e-mask can no longer be used or the integrity of the e-mask has been broken?
  5. According to Figure 1, only a small part of the mask was covered by the printable PEDOT: PSS ink. Does that mean only part integrity of the mask can be monitored? What will happen if there a small hole in other parts of the mask other than the part covered by the ink? The authors may do some experiments to demonstrate these kind of situations.

Author Response

Please see our point by point response in the doc

Round 2

Reviewer 1 Report

No further comment

Author Response

ok

Reviewer 2 Report

The authors have answered all the questions and revised the manuscript accordingly.

Author Response

ok